# Optimization of Anti-Plugging Working Parameters for Alternating Injection Wells of Carbon Dioxide and Water

Kemin Li [1], Guangsheng Cao [2,*], Gaojun Shan [3], Ning Zhang [2], Xincheng Liu [2], Shengbo Zhai [2] and Yujie Bai [2]

[1] Quality, Safety and Environmental Protection Supervision and Evaluation Center of Daqing Oilfield Co., Ltd., Daqing 163000, China

[2] Key Laboratory of Enhanced Oil & Gas Recovery of Ministry of Education, Northeast Petroleum University, Daqing 163318, China

[3] Exploration and Development Research Institute, Daqing Oilfield Co., Ltd., Daqing 163000, China

* Correspondence: nepucgswz@163.com

**Abstract:** In the process of oilfield development, the use of $CO_2$ can improve the degree of reservoir production. Usually, $CO_2$ is injected alternately with water to expand the spread range of $CO_2$, and $CO_2$ presents a supercritical state in the formation conditions. In the process of alternating $CO_2$ and water injection, wellbore freezing and plugging frequently occur. In order to determine the cause of freezing and plugging of injection wells, the supercritical $CO_2$ flooding test area of YSL Oilfield in China is taken as an example to analyze the situation of freezing and plugging wells in the test area. The reasons for hydrate freezing and plugging are obtained, the distribution characteristics and sources of hydrate near the well are clarified, and a coupling model is established to calculate the limit injection velocity and limit shut-in time of $CO_2$ and water alternate injection wells. The results show that the main reasons for freezing and plugging of supercritical $CO_2$ water alternate injection wells are long time shut down after alternate injection, improper operation when stopping injection and starting and stopping pumps, and slow injection speed during alternate injection. In the process of supercritical $CO_2$ water alternative injection, in the case of post-injection, the $CO_2$ in the formation will reverse diffuse to the injection well end. With the continuous increase of daily water injection, the initial diffusion position and the time of $CO_2$ diffusion to the perforated hole after well shut-in gradually increase. The time of $CO_2$ reverse diffusion to the bottom of the well is 1.6–32.3 d, and the diffusion time in the perforated hole is 1.0–4.5 d. Therefore, the limit shut-in time following injection is 2.6–36.8 d. Following gas injection, the limit shut-in time of a waterproof compound can be divided into three stages according to the change of wellbore pressure: the pressure stabilization stage, pressure-drop stage and formation fluid-return stage. The limit shut-in time of a waterproof compound following gas injection is mainly affected by permeability, cumulative gas injection rate and formation depth. The limit shut-in time of a waterproof compound is 20.0~30.0 days. The research results provide technical support for the wide application of $CO_2$ flooding.

**Keywords:** alternate injection of $CO_2$ and water; hydrate freeze plugging; limit shut-in time

---

## 1. Introduction

For oilfield development, $CO_2$ displacement is adopted to improve oil and gas recovery. However, during the injection process, $CO_2$ hydrate will be generated in the wellbore of the injection well due to the injection of hydrocolloids, and $CO_2$ hydrate will freeze and block the wellbore. Davy et al. [1] discovered chlorine hydrate in the laboratory for the first time. At the time, it was generally believed that the hydrate was a special inorganic compound. De Forcrand et al. [2] measured the equilibrium temperatures of 15 different hydrates through indoor hydrate synthesis experiments, including the equilibrium temperatures of natural gas and $CO_2$ hydrates under environmental pressure (100 kPa). Hammerschmidt [3] determined that there will be gas hydrate formation in the pipeline at freezing temperatures

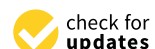



through $CO_2$ and methane hydrate formation experiments, which will lead to pipeline blockage and affect the stability of the fluid flow process. Claussen et al. [4] found that the hydrate is a three-dimensional structure through X-ray diffraction and observed that the structure's surface is pentagonal. Then, comparing the size of the methane molecule with the internal space size of a pentagonal dodecahedral water molecule, they found that the internal space can accommodate guest molecules and proposed the molecular structure of the hydrate, that is, the S-I cubic crystal structure. Von Stackelberg et al. [5] confirmed this structure by X-ray diffraction of hydrate. Claussen et al. [6] proposed another hydrate structure, the S-II cubic crystal structure, through X-ray diffraction of chloride hydrate and calculation of the molecular energy value, which marked the beginning of studies of the microstructure and macroproperties of hydrate. Makogon et al. and Davidson et al. [7] studied hydrate by molecular dynamics simulation for the first time. Ripmeester et al. [8] found a new structure of hydrate, namely the S-H hexagonal crystal structure, by using spectral diffraction and nuclear magnetic resonance methods. S-I, S-II, and S-H structures are considered to be the three most common hydrocarbon hydrate crystal structures. Svartaas [9] preliminarily obtained the nucleation rate of $CH_4$-$H_2O$ hydrate by applying the PDF method, which has been recognized as applicable. At the same time, Maeda [10] measured the nucleation rate of natural gas hydrate, obtained the nucleation curve of natural gas hydrate, and believed that the nucleation rate was linear with the system size through subsequent research [11]. Amadeu K. [12] et al. studied the accelerator of hydrate formation and believed that under the condition of a high shear rate, a large amount of hydrate crystal nuclear energy was generated, which could increase the macro-formation rate of hydrate. In the process of carbon dioxide displacement of crude oil, the interface characteristics of carbon dioxide hydrate in the injected string have a greater impact on the freezing of and plugging by hydrate. Luis E. [13] et al. established a multiphase coupling model based on a simple slug flow model. Ngoc N [14] et al. studied the interaction between the surface premelting and the interface of hydrate and studied the interface characteristics of hydrate formation through molecular simulation. Sloan [15] conducted a detailed study on the nucleation characteristics of hydrate and analyzed the influencing factors for hydrate nucleation. Moon [16] et al. conducted a numerical simulation study on the characteristics of hydrate in the wellbore during the development of deep-water gas fields and established a prediction model for hydrate plugging.

Although many scholars have done a lot of research on the formation and decomposition characteristics of $CO_2$ flooding and hydration, the reasons for the freezing and plugging of supercritical $CO_2$ and water alternate injection wells are still unclear. The formation pressure of the $CO_2$ drive oil wells in the Daqing YSL Oilfield is high, up to 35 mpa, and the temperature is also high, about 90 °C, so carbon dioxide hydrate cannot be generated in the formation. However, for injection wells, the temperature in the wellbore of the injection well is relatively low, the wellhead temperature can reach 10 °C in winter, and the wellhead injection pressure can reach 20 mpa. Therefore, when water and carbon dioxide exist near the wellhead of the injection well, it is possible to generate hydration and cause hydration blockage. In addition, the preventive measures of temperature on hydrate freeze plugging are not perfect. According to the actual conditions of hydrate plugging wells in the YSL Oilfield, hydrate plugging wells can be divided into two categories: plugging during water injection and plugging during well shut-in. Taking the supercritical $CO_2$ flooding test area of YSL Oilfield in China as an example, this paper analyzes the situation of frozen plugging wells in the test area, classifies the reasons for hydrate plugging of wellbores, clarifies the distribution characteristics and sources of hydrate near the wellbores, establishes a coupling model to calculate the limit injection velocity and limit shut-in time of $CO_2$ and water alternate injection wells, and provides technical support for the wide application of $CO_2$ flooding.

## 2. Methodology

### 2.1. Analysis of Freezing and Plugging in Supercritical CO<sub>2</sub> and Water in Alternate Injection Wells

*2.1. Analysis of Freezing and Plugging in Supercritical $CO_2$ and Water in Alternate Injection Wells*

Due to the low productivity of low-permeability reservoirs, fracturing development is generally adopted to ensure the productivity of oil wells and the injection capacity of water wells. The artificial fractures and fractures are interlaced near the well. The injected fluid enters the formation from the wellbore along the fractures at a fast flow rate, while the seepage velocity in the rock matrix is low. At the same time, due to the fluidity and adsorption performance of supercritical $CO_2$ under the formation conditions, $CO_2$ will remain near the wellbore of the formation when injecting $CO_2$ slugs, and the residual $CO_2$ will reverse-diffuse along the low-flow-rate area and then enter the perforating hole. Since the flow in the perforated hole of the injection well is variable-mass flow, the flow velocity near the top of the perforation is high, and the flow velocity in the hole at the bottom of the perforation is low. In addition, due to the viscous flow of the injected water, the flow velocity near the perforated hole wall is low. [17] Therefore, after $CO_2$ enters the perforated hole, it will further reverse diffusion along the low-flow-rate perforated hole and the hole wall and enter the wellbore. At this time, the $CO_2$ entering the wellbore is divided into two parts. One part continues to migrate upward along the low-velocity area of viscous flow, namely the pipe wall of the water injection pipe string, to reach the well section meeting the hydrate formation conditions and the frozen block section; the other part gradually gathers to form bubbles, which rise to the freezing block section by buoyancy, and finally form hydrate. In the process of alternative supercritical $CO_2$ and water injection, under the condition of water injection before and after well shut-in, the source of $CO_2$ in the hydrate in the wellbore freeze plugging section is mainly the $CO_2$ slug near the wellbore. The analysis showed that the position of the $CO_2$ slug is affected by the linear velocity of injected water in the formation, and the process is shown in the Figure 1.

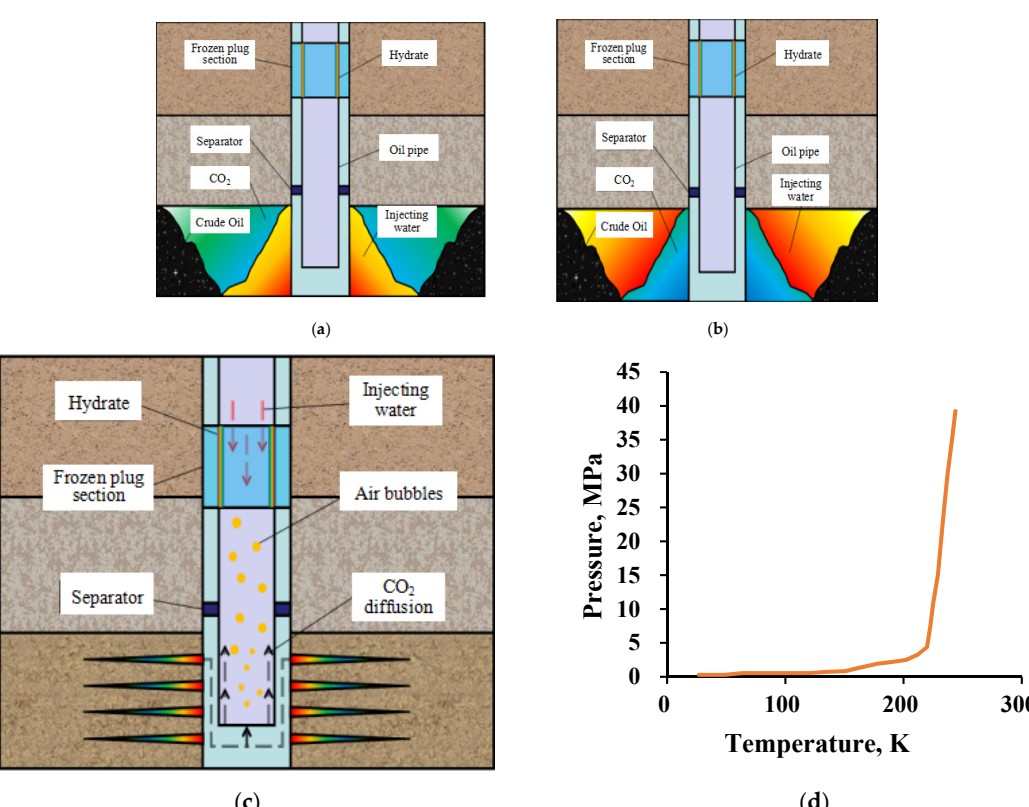

**Figure 1.** Classification of hydrate plugging in supercritical $CO_2$ and water alternative injection wells, and $CO_2$ phase equilibrium curve. (**a**) During water injection before and after well shut-in. (**b**) During gas injection before and after shut-in. (**c**) During normal water injection. (**d**) Carbon dioxide phase equilibrium curve.

Following injection, the $CO_2$ that forms carbonate hydrate in the wellbore after shut-in comes mainly from the supercritical $CO_2$ slug in the formation [18]. Therefore, the limit shut-in time at this time can be calculated according to the time when the $CO_2$ in the formation enters the wellbore. This process is mainly based on the diffusion of $CO_2$ in the water. In the case of $CO_2$ injection before and after the well was shut in, the water in the hydrate in the wellbore freeze plugging section mainly came from the previously injected water slug. After the well is shut in, the phase state of the remaining $CO_2$ in the wellbore gradually changes from the liquid phase to the gas phase, and the $CO_2$ dissolves in the water and further diffuses into the formation [19]. At the same time, to ensure the dynamic balance between the bottom hole pressure and the formation pressure, the formation water will gather in the wellbore. When the water enters the location of the freeze-plugging section, the hydrate freeze-plugging can occur. For the calculation of the limit shut-in time in the wellbore after shut in following gas injection, it is necessary to focus on the source of water in the wellbore frozen plugging section. While the pressure in the wellbore gradually decreases during the water return process, the water can only rise to the frozen plugging section by relying on the boundary pressure. Therefore, the rise rule for injected water in the formation can be studied by studying the change in pressure in the wellbore after shutting in, and then the limit shut-in time can be calculated.

### 2.2. Establishment of a Reverse Diffusion Model in Wellbore and Formation

According to the analysis of the source of $CO_2$ in the wellbore during water injection, the fluid flow, material diffusion, carrying force of the fluid on the bubbles and the buoyancy of the bubbles themselves should be considered to determine the source of $CO_2$ [20].

(1) Fluid flow in the wellbore

$$\rho \frac{\partial u}{\partial t} + \rho(u \cdot \nabla)u = \nabla \cdot \left\{ -p_I + \mu \left[ \nabla \mu + (\nabla \mu)^T \right] \right\} + F + \rho g \tag{1}$$

where $\rho$ Is the fluid density, $kg/m^3$; $u$ is the fluid velocity in the wellbore, m/s; $\mu$ is the viscosity of injected water, Pa·s; $t$ is dimensionless time; $p_I$ is the fluid pressure in the perforation hole, Pa; $T$ is the temperature, K; $F$ is the volume force on the fluid, $N/m^3$; $g$ is the acceleration of gravity, $m/s^2$.

(2) Diffusion of $CO_2$ in the wellbore

$$\frac{\partial c_i}{\partial t} + \nabla \cdot (-D_i \nabla c_i) + u \cdot \nabla c_i = R_i \tag{2}$$

$$N_i = -D_i \nabla c_i + u c_i \tag{3}$$

where $c_i$ is the diffusion coefficient, $m^2/s$; $D_i$ is the initial concentration of microelement segment, $mol/m^3$; $\nabla$ is Hamiltonian operator; $R_i$ is the concentration at the end of the microelement segment, $mol/m^3$.

(3) Carrying force of fluid on $CO_2$ bubble

$$F_D = \frac{1}{\tau_P} m_P (u' - v) \tag{4}$$

where $F_D$ is the carrying force of a fluid on bubbles, N; $\tau_P$ is the corresponding time of bubble velocity, m/s; $m_P$ is the bubble mass, kg; $u'$ is the fluid velocity in the wellbore, m/s; $v$ is the bubble velocity, m/s;

Including:

$$\tau_P = \frac{\rho_P d_P{}^2}{18\mu} \tag{5}$$

where $\rho_P$ is bubble density, $kg/m^3$; $d_P$ is the bubble diameter, m.

(4) Buoyancy of $CO_2$ bubbles [20]

$$F_g = m_p g \frac{\rho_p - \rho}{\rho_p} \tag{6}$$

where $F_g$ is the buoyancy of the bubble, N; $\rho$ is water density, kg/m$^3$.

The source of $CO_2$ in the frozen section is mainly affected by the reverse diffusion of $CO_2$ near the well. According to the fluid flow diffusion model in the wellbore and formation, combined with the actual construction data, the fluid flow and $CO_2$ reverse diffusion process in the wellbore and formation are analyzed to provide a basis for subsequent process design and calculation of limit injection parameters. According to the actual construction data and formation data of $CO_2$ flooding in the YSL Oilfield, the design simulation parameters are shown in Table 1.

**Table 1.** Parameters required for simulation.

| Project | Data | Company | Project | Data | Company |
|---|---|---|---|---|---|
| daily water injection | 20 | m$^3$/d | perforating depth | 0.5 | m |
| general injection wellbore diameter | 40 | mm | perforation density | 24 | hole/m |
| diameter of double-pipe injection side pipe | 40 | mm | formation pressure | 10 | MPa |
| casing diameter | 139 | mm | formation permeability | 20 | $10^{-3}$ μm$^2$ |
| perforating hole diameter | 12 | mm | diffusion coefficient | 2.84 | $10^{-8}$ m$^2$/s |

*2.3. Induction Time of Hydrate Formation in Wellbore during $CO_2$ Injection before and after Shut-In*

Under the condition of post-water-injection for gas injection wells, the frozen plugging section in the wellbore meets the hydrate generation conditions. The limiting shut-in time of post-water-injection wells is the time when the $CO_2$ in the wellbore diffuses from the formation to the frozen plugging section of the wellbore. However, under the condition of post-water-injection for gas injection wells, the bottom hole pressure of gas injection wells is high, and free $CO_2$ will not return to the gas injection wells; however, $CO_2$ will be dissolved in water. Therefore, the dissolution of $CO_2$ in formation water cannot be ignored for the calculation of the limit shut-in time following water injection [21]. Therefore, the following three conditions need to be considered for the calculation of the limit shut-in time following water injection.

① During water injection, whether the $CO_2$ in a certain position in the formation has reverse diffusion; that is, whether under the condition of normal injection of the injection well, the $CO_2$ in the formation will diffuse to the injection well end. If reverse diffusion occurs and the diffusion speed is fast, it is unnecessary to consider the diffusion speed of $CO_2$ in the formation, and only to calculate the time when the $CO_2$ in the wellbore diffuses from the bottom of the well to the frozen position after the well is shut in.

② Because the injected water flows radially from the injection well to the production well, the injection rate of the injected water in the formation is slow. Near the wellbore zone, due to the impact of perforations, reverse diffusion will occur at the bottom perforated hole wall at a low flow rate. The residual $CO_2$ in the formation porous media will gradually diffuse into the wellbore. After the local formation $CO_2$ enters the wellbore under the influence of gravity, the ultra-critical $CO_2$ entering the wellbore will migrate upward. At this time, under the influence of temperature and pressure in the wellbore, the supercritical $CO_2$ will undergo phase change and become gaseous. The diffusion time of $CO_2$ from the formation to the wellbore along the perforated hole can be simulated by the finite element method. The rise time of $CO_2$ entering the wellbore along the wellbore is shorter than the diffusion time in the formation and hydrate generation time, so it can be ignored.

③ When the $CO_2$ in the wellbore diffuses to the freezing and blocking position, $CO_2$ hydrate will be generated in the wellbore under the appropriate temperature and pressure.

The formation time of $CO_2$ hydrate is the induction time of hydrate formation, which can be determined through indoor experiments.

$$t_{PWI} = t_{DIFF} \times 60 \times 60 + t_{IND} \tag{7}$$

where $t_{PWI}$ is the limit shut-in time of hydrate freezing in the wellbore during post $CO_2$ injection, s; $t_{DIFF}$ is the time when $CO_2$ enters the bottom hole from the formation, h; and $t_{Ind}$ is the induction time of hydrate formation, s.

The phase change process of fluid in the wellbore of the $CO_2$ injection well after shut-in can be divided into the following three stages: (1) pressure stabilization stage; (2) pressure-drop stage; (3) formation fluid reflux stage.

When water and $CO_2$ exist simultaneously in the frozen plugging section of the wellbore, the conditions for hydrate formation are met. Under this condition, the time from the beginning of $CO_2$ hydrate formation to the complete plugging of the pipeline is the induction time of hydrate formation. Then the limit shut-in time of hydrate formation in the frozen plugging section following gas injection can be expressed as:

$$t_{PGI} = t_{STA} \times 60 \times 60 \times 24 + t_{DES} + t_{BLO} + t_{IND} \times 60 \tag{8}$$

where $t_{PGI}$ is the limit shut-in time of hydrate freezing in the wellbore following $CO_2$ injection, s; $t_{STA}$ is the stabilization time of bottom hole pressure, s; $t_{DES}$ is the time of bottom hole pressure drop, s; $t_{BLO}$ is the time of formation water flowing back to the wellbore freezing and plugging section, s; and $t_{Ind}$ is the induction time of hydrate formation, min.

(1) Pressure stabilization stage

The residual liquid $CO_2$ in the wellbore is converted into supercritical $CO_2$. Due to the compressibility of the gas, it is considered that the wellbore and bottom hole pressures are unchanged at this stage, and $CO_2$ is still injected into the formation in a supercritical state until it is completely injected into the formation. The time in this process is related to the wellbore size, original injection flow rate, and the depth of the supercritical $CO_2$ critical temperature point as follows:

$$t_{STA} = \frac{\pi D^2 L}{4q} \times 10^{-6} \tag{9}$$

where $D$ is the diameter of the tubing, mm; $L$ is the depth of the downhole freezing point, m; and $q$ is the flow during normal injection, m$^3$/d.

(2) Pressure-drop stage

When the residual liquid $CO_2$ is gasified, the bottom hole pressure of the gas injection well will conform to the change characteristics of the bottom hole pressure after the shut in of the injection well, while the bottom hole pressure and temperature of the conventional injection well meet the conditions of the supercritical state of $CO_2$. While the supercritical $CO_2$ is between the liquid state and the gaseous state, its density is similar to the liquid state, its viscosity is similar to the gaseous state, and the fluid compressibility is low. Therefore, the change of bottom hole pressure following gas injection should follow the rule of bottom hole pressure change for water injection wells, and the viscosity of gaseous $CO_2$ is selected as the fluid viscosity, so the rule of bottom hole pressure change during the pressure-drop stage after the shutting-in of supercritical $CO_2$ gas injection wells meets the Horner formula:

$$P_{\text{ws}}(\Delta t) = P_{\text{wf}}(\Delta t = 0) + 0.183 \frac{q\mu}{Kh} \lg \frac{2.25\eta \Delta t}{r_{\text{we}}^2} \tag{10}$$

where $P_{\text{ws}}$ is the bottom hole pressure after shut-in time t, Pa; $P_{\text{wf}}$ is the instantaneous bottom hole pressure during shut in, Pa; $h$ is the thickness of the oil layer, m; $\eta$ is the formation conductivity coefficient, m$^2$/s; and $r_{\text{we}}$ is the effective radius of the wellbore, m.

According to Formulas (2)–(14), the time of the pressure-drop stage can be obtained $t_{DES}$:

$$t_{DES} = \frac{r_{we}{}^2 \times 10^{\frac{Kh\ (P_{wf}\ (\Delta t=0)\ -P_{ws}\ (\Delta t))}{0.183q\mu}}}{2.25\eta} \tag{11}$$

where $W_t$ is the cumulative injection volume, m³; $E_v$ is the sweep coefficient, dimensionless; $P_e$ is the formation pressure, MPa; $P_{wf}$ is the bottom hole pressure formed by casing pressure and gas column in the wellbore, MPa; and $r_e$ is the radius of the supply edge, m.

(3) Formation fluid-return stage

When the pressure in the wellbore decreases to a certain extent, the bottom hole pressure and the formation pressure reach equilibrium. At this time, there is pure gaseous and supercritical $CO_2$ in the wellbore, and $CO_2$ will dissolve and diffuse into the formation fluid, causing the pressure in the wellbore to gradually decrease. The balance between the bottom hole pressure and the formation pressure cannot be maintained for a long time, depending on the casing pressure and the liquid column pressure of multiphase fluid in the wellbore. Therefore, in order to maintain the balance between the bottom hole pressure and the formation pressure, the formation water level will rise in the wellbore, and hydrate will be formed when the liquid in the wellbore rises to the frozen plugging section. However, the formation return also involves the subsequent gas injection at the end of the previous water injection slug. When the subsequent gas injection slug volume is large, the return time of the formation fluid is long, so the time of the formation fluid return phase $t_{BLO}$ can be expressed as:

$$t_{BLO} = \frac{W_t\mu \ln\left(\frac{r_e}{r_w}\right)}{Kh\ (P_e - P_{wf})} \cdot E_V \tag{12}$$

## 3. Results and Discussions

### 3.1. Calculation of the Lowest Flow Rate in Injection Well during Water Injection

According to the flow law in the wellbore, perforating hole and formation during water injection, it is judged that the reverse diffusion of $CO_2$ will start near the wellbore and enter the wellbore through the perforating hole; some $CO_2$ will enter the wellbore and gradually gather to form bubbles, and the other part will continue to reverse-diffuse along the pipe wall. As the reverse diffusion of $CO_2$ cannot be avoided, and the amount of reverse diffusion is small, it takes a long time to completely freeze and block, so it is only necessary to optimize the process parameters for the gathered $CO_2$ bubbles.

In order to clearly describe the migration law of bubbles in the injection well during water injection, combined with the data in Table 1, the $CO_2$ bubble inlet is set as the bottom of the model, the outlet is set as the top of the model, the injection water inlet and outlet are opposite to the bubble movement direction, the bubble release cycle is 50 min per piece, the release time is 3000 min, the water viscosity is 0.5 mPa/s, and the bubble density is 1.816 kg/m³. The rising rule of bubbles in injection wells during water injection is simulated as shown in Figure 2.

It can be seen from Figure 2 that the bubble-carrying capacity of injected water is limited, and the reverse migration of $CO_2$ bubbles cannot be inhibited under the simulation conditions described in Table 1, which indicates that the current daily water injection volume in the $CO_2$ flooding test area of YSL Oilfield is not reasonable, and $CO_2$ bubbles can migrate upward to the frozen plugging section to cause frozen plugging. Hydrate generation can be inhibited by changing the water injection flow rate. Increasing the fluid flow rate can enhance the drag force of injected water on $CO_2$ bubbles. Under simulation conditions, when the daily water injection volume reaches 360 m³/d, the number of $CO_2$ bubbles in the wellbore is 1. If the water injection intensity continues to increase, the number of bubbles in the wellbore will become 0, indicating that an appropriate amount of daily water injection can effectively control the upward migration of $CO_2$ bubbles and inhibit the formation of hydrate in the frozen section during water injection. It is determined that

the limit injection volume to prevent bubbles from migrating upward under simulation conditions is 400 m³/d. According to the limit injection volume and formula (13), the injection water velocity preventing bubbles from migrating upward can be calculated:

$$v = \frac{4 \times 1000^2 Q}{24 \times 60 \times 60 \pi D^2} \tag{13}$$

where $v$ is the minimum flow rate of injected water to prevent $CO_2$ bubbles from moving upward, m/s; $Q$ is the daily water injection volume, m³/d; and $D$ is the wellbore diameter of injection well, mm.

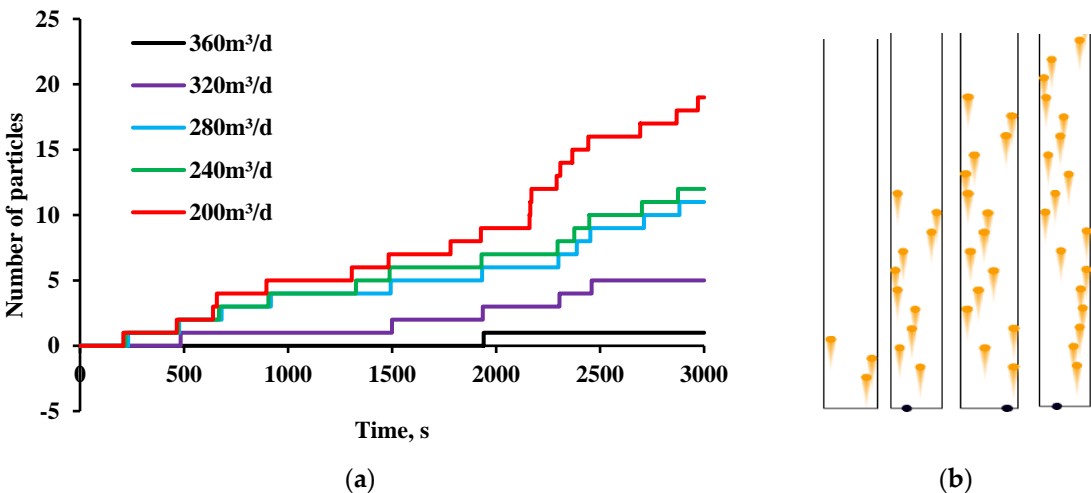

**Figure 2.** Number of $CO_2$ bubbles in wellbore under different daily injection−water quantities.(**a**) Particle number change; (**b**) Particle distribution state.

The minimum size of the throttle valve port in the throttle under low injection intensity is calculated according to Formula 13, as shown in Table 2.

**Table 2.** Choke valve port size selection under low injection intensity.

| Serial No | Daily Injection Volume (m³/d) | Tubing Size (mm) | Flow Rate in Tubing (m/s) | Limiting Velocity (m/s) | Throttle Valve Port Size (mm) |
|---|---|---|---|---|---|
| 1 | 5 | | 0.046 | | 5.44 |
| 2 | 10 | | 0.092 | | 7.70 |
| 3 | 15 | | 0.138 | | 9.42 |
| 4 | 20 | 40 | 0.184 | 1.53 | 10.88 |
| 5 | 25 | | 0.230 | | 12.17 |
| 6 | 30 | | 0.276 | | 13.33 |

It can be seen from Table 2 that the limit flow rate to prevent carbon dioxide from migrating upward in the wellbore is 1.53 m/s. Under the condition of a low daily injection rate, the flow rate in the tubing ranges from 0.046 m/s to 0.276 m/s, which cannot reach the limit flow rate; according to the calculation of the limit flow rate, the maximum size range of the throttle valve port in the throttle is 5.44 mm~13.33 mm.

### 3.2. Limit Shut-in Time of Injection Well Waterproof Compound during $CO_2$ Injection before and after Shut In

Based on the basic data, the fluid flow model in porous media and the multi-physical field coupling model of dilute material transfer are established, and numerical simulation is carried out to analyze and judge whether the reverse diffusion of $CO_2$ will occur in the

formation. In addition, during the normal injection process, the $CO_2$ adsorbed on the rock pore wall in the formation will not diffuse into the wellbore, but when the injection is stopped and the well is shut in, the $CO_2$ in the formation will diffuse into the wellbore; $CO_2$ hydrate will be generated when $CO_2$ entering the wellbore floats up to the wellbore freezing position due to the influence of gravity and the wellbore temperature and pressure. In order to determine the time when $CO_2$ diffuses from the formation to the bottom of the well through the perforation hole, the perforation hole diameter is set at 12 mm and the perforation depth is 0.5 m. As the diffusion speed of $CO_2$ in the perforated hole is the same, the number of perforations has no effect on the diffusion of $CO_2$. For convenience, the number of perforations is selected as eight. Considering the conditions of fluid flow in the wellbore, combined with the rare material transfer model, a multi-physical field coupling diffusion model of $CO_2$ in the wellbore from bottom to top is established, and the diffusion process of $CO_2$ in the wellbore and formation is simulated as shown in Figure 3.

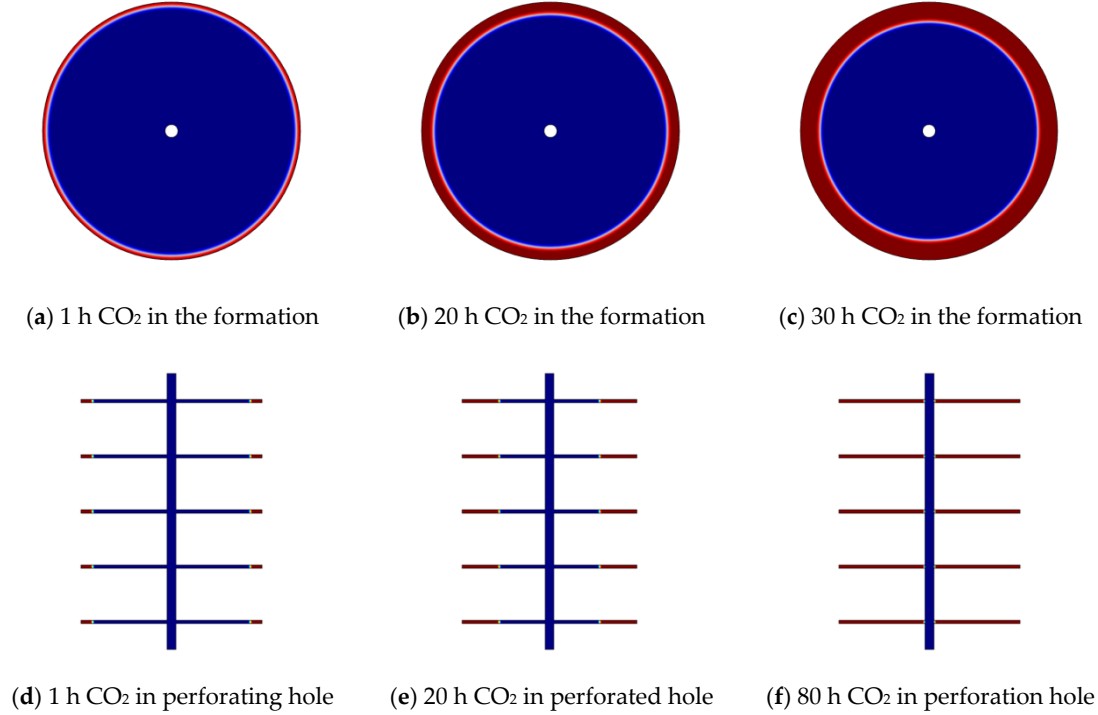

(**a**) 1 h $CO_2$ in the formation   (**b**) 20 h $CO_2$ in the formation   (**c**) 30 h $CO_2$ in the formation

(**d**) 1 h $CO_2$ in perforating hole   (**e**) 20 h $CO_2$ in perforated hole   (**f**) 80 h $CO_2$ in perforation hole

**Figure 3.** Diffusion process of $CO_2$ in wellbore and formation.

It can be seen from the simulation results that, due to the slow percolation speed of the injected water in the formation, even under the condition of normal injection, the $CO_2$ in the formation will also reverse diffuse to the injection well end. In addition, before the alternative water injection, the $CO_2$ in the formation is in a supercritical state, and the $CO_2$ in this state is more likely to be adsorbed on the wall of the formation rock. During the subsequent water injection process, $CO_2$ will always remain in the formation near the well, so it is unnecessary to consider the reverse diffusion of $CO_2$ in the formation.

When the shut-in time reaches 80 h, the $CO_2$ in the formation will enter the wellbore along the perforation hole, which is to say, without considering the induction time of $CO_2$ hydrate formation and the time of $CO_2$ rising in the wellbore, the limit shut-in time for post-water-injection should not be higher than 80 h. According to the simulation results, the limit shut-in time of water repellent compound freeze plugging following water injection mainly includes the induction time of hydrate formation, the time of $CO_2$ diffusion to the bottom of the well and the time of $CO_2$ rising to the freeze plugging section under the action of buoyancy. Due to the relatively short time of $CO_2$ rising to the freezing block section due to buoyancy and the induction time of hydrate formation, it can be ignored. Although there are slight differences in hydrate formation under different temperatures and pressures,

the actual shut-in time of the oilfield is generally more than one day. Therefore, the main factor affecting $CO_2$ hydrate formation after shut in of gas injection wells is the time of $CO_2$ diffusion to the bottom of the well. The diffusion process of $CO_2$ from the formation to the wellbore includes diffusion of $CO_2$ in the porous medium and in the perforated hole, that is, the influence of initial diffusion position and diffusion speed. The initial diffusion position is affected by the daily water injection rate, and the diffusion speed is affected by the formation temperature. The changes of formation temperature and diffusion starting position under different daily water injection rates and the time changes of $CO_2$ diffusion to perforated hole after well shut-in are simulated as shown in Figure 4.

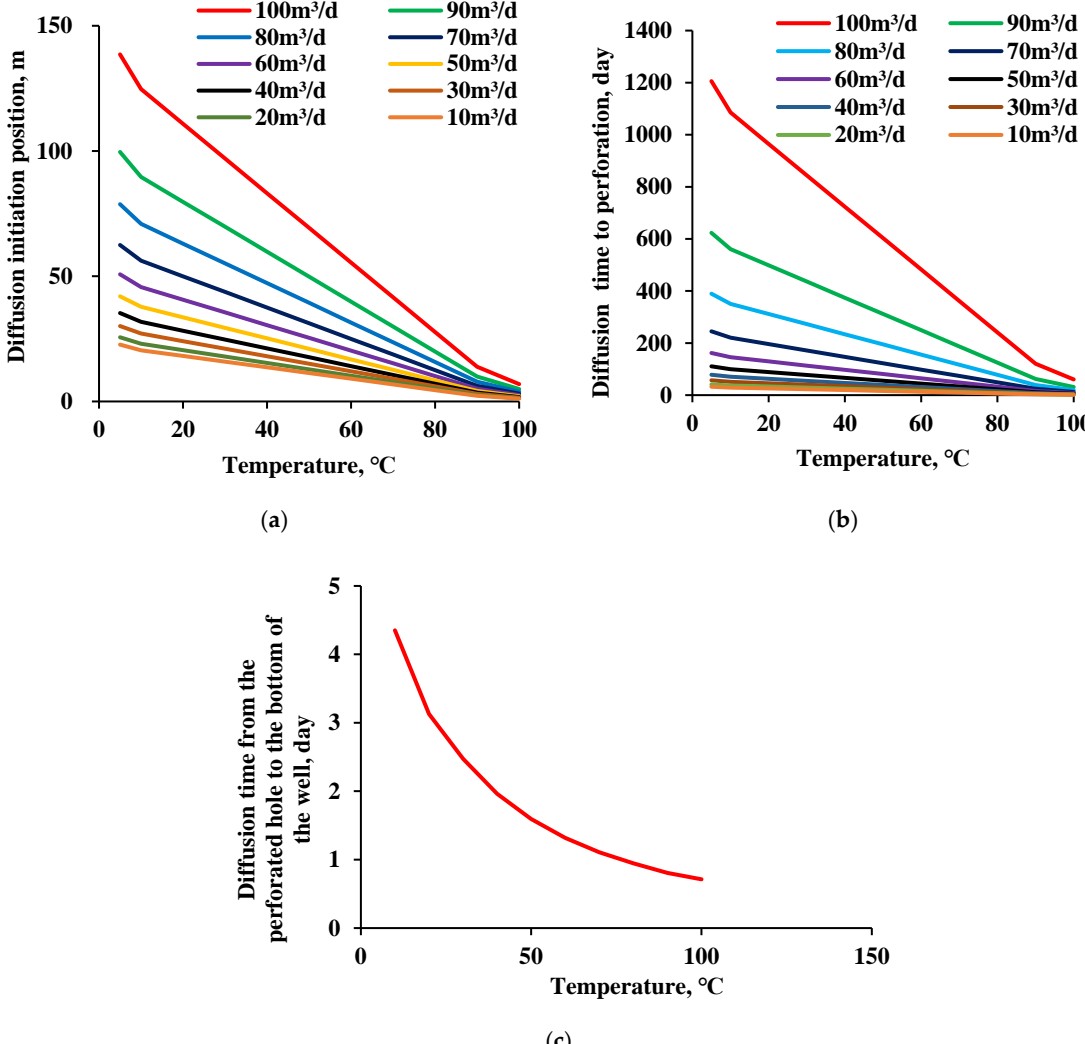

**Figure 4.** Diffusion time change of $CO_2$ in stratum under different conditions. (**a**) Variation of formation temperature and initial diffusion position under different daily water injections; (**b**) Variation of formation temperature and diffusion time to perforated hole under different daily water injection; (**c**) Diffusion time at different temperatures.

It can be seen that with the continuous increase of daily water injection, the initial diffusion position and the time of $CO_2$ diffusion to the perforation hole after well shut-in gradually increase. Taking the water gas alternative $CO_2$ injection well in the YSL Oilfield as an example, the formation temperature is 90 °C, and the time of $CO_2$ gas diffusion to the perforation hole position in the formation is about 1.6–32.3 d when the daily water injection is 5 $m^3$/d–100 $m^3$/d. The diffusion time in the perforated hole is generally about 1–4.5 d. The temperature has a great influence on the diffusion time of $CO_2$ from the perforated hole to the bottom hole. With the increase in formation temperature, the diffusion time to

the bottom hole gradually decreases. The impact of daily water injection on the limiting shut-in time of a waterproof compound formation is mainly reflected in the initial position of diffusion during shut-in. When the daily water injection is high, the seepage velocity in the formation is relatively fast, and the fluid flow velocity in the formation will gradually decrease with the increase in displacement distance due to the action of radial flow. When the fluid flow velocity in the local formation is higher than the diffusion velocity of $CO_2$ gas in the formation, the position from this point to the water injection well is the initial position of $CO_2$ reverse diffusion when the well is shut in.

### 3.3. Limit Shut-in Time of Hydrate Formation in Wellbore during Water Injection after Shutting In

The limit shut-in time to prevent $CO_2$ hydrate formation following gas injection can be characterized by the change of bottom hole pressure after shut-in. When the bottom hole pressure and formation pressure reach equilibrium, the formation water will enter the wellbore. Therefore, the wellbore size, injection flow, depth of the supercritical $CO_2$ critical temperature point, formation factor, cumulative injection volume, etc., all affect the limit shut-in time for hydrate formation following gas injection, as shown in Figure 5.

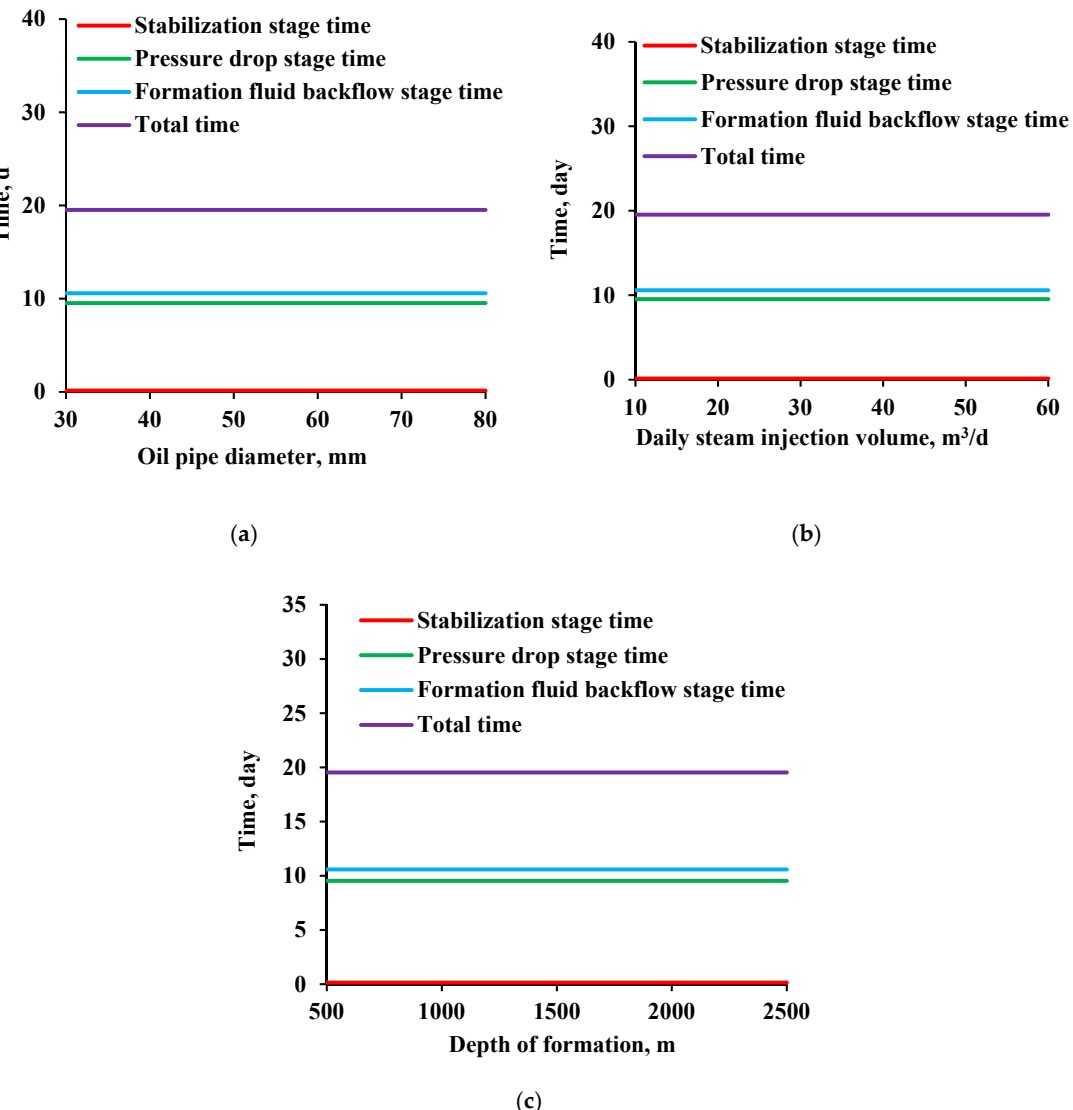

**Figure 5.** Influence of well bore size, daily gas injection rate and critical depth on the limit shut-in time of a waterproof compound. (**a**) Influence of tubing diameter on the limit shut-in time; (**b**) Influence of daily gas injection rate on limiting shut-in time; (**c**) Influence of critical depth of liquid $CO_2$ on the limit shut-in time.

By comparing the time changes of the stabilization stage, bottom hole pressure-drop stage and formation fluid return stage, it can be seen that the stabilization stage is relatively short, while the time of the formation fluid-return stage and pressure-drop stage is relatively long. Therefore, the limit shut-in time of the waterproof compound following gas injection should be mainly affected by the time of the pressure-drop stage and formation fluid return stage, while the time of the stabilization stage has little impact. Therefore, the corresponding well bore size, daily gas injection rate and critical depth have little influence on the limiting shut-in time.

As the pressure-drop stage and the formation fluid return stage involve the conduction of fluid in the formation through the porous medium, the permeability of the formation rock and the thickness of the oil layer have a certain influence on the limit shut-in time of the waterproof compound. The influence of the formation permeability and of the thickness of the oil layer at different stages on the limit shut-in time of the waterproof compound is calculated as shown in Figure 6.

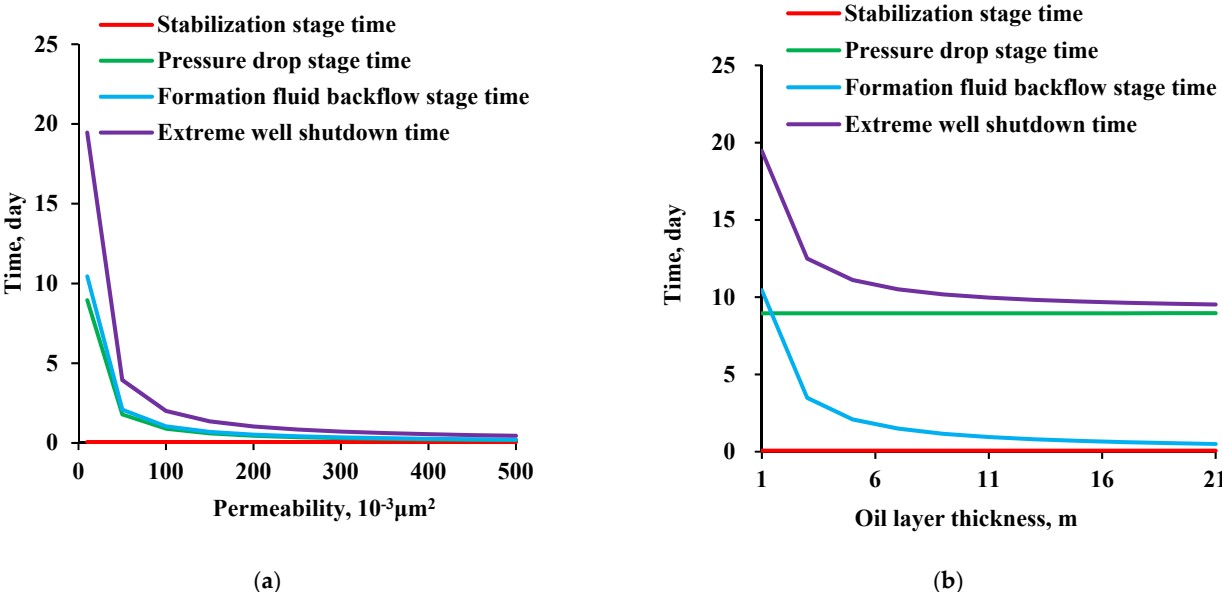

(**a**)　　　　　　　　　　　　　　　　　　　　　　　　　　　　　　　　　　　(**b**)

**Figure 6.** Effect of formation permeability and reservoir thickness on the limit shut-in time of water proof compound. (**a**) Influence of formation permeability on the limit shut-in time of water proof compound; (**b**) Influence of reservoir thickness on the limit shut-in time of water proof compound.

It can be seen that formation permeability and reservoir thickness have a great influence on the limit shut-in time of a waterproof compound following gas injection. The influence of formation permeability on the limit shut-in time of a waterproof compound is mainly reflected in the pressure-drop stage and the formation fluid backflow stage and has the greatest influence on the limit shut-in time of a waterproof compound following the gas injection. However, the influence of reservoir thickness on the limit shut-in time of a waterproof compound is mainly reflected in the stage of formation fluid backflow. The analysis shows that the influence of formation permeability on the limit shut-in time of a waterproof compound is mainly from the aspects of pressure recovery time and pressure transmission capacity, and the influence of reservoir thickness on pressure recovery and transmission is not significant.

The influence of reservoir thickness on the ultimate shut-in time of the waterproof compound is mainly reflected in the initial diffusion position of $CO_2$ during shut-in, so the cumulative gas injection rate should also have a greater impact on the ultimate shut-in time of the waterproof compound. In addition, considering that the bottom hole pressure in the flowback stage is mainly affected by the formation depth, the influence of the cumulative

gas injection rate and the formation depth on the limit shut-in time of the waterproof compound is calculated as shown in Figure 7.

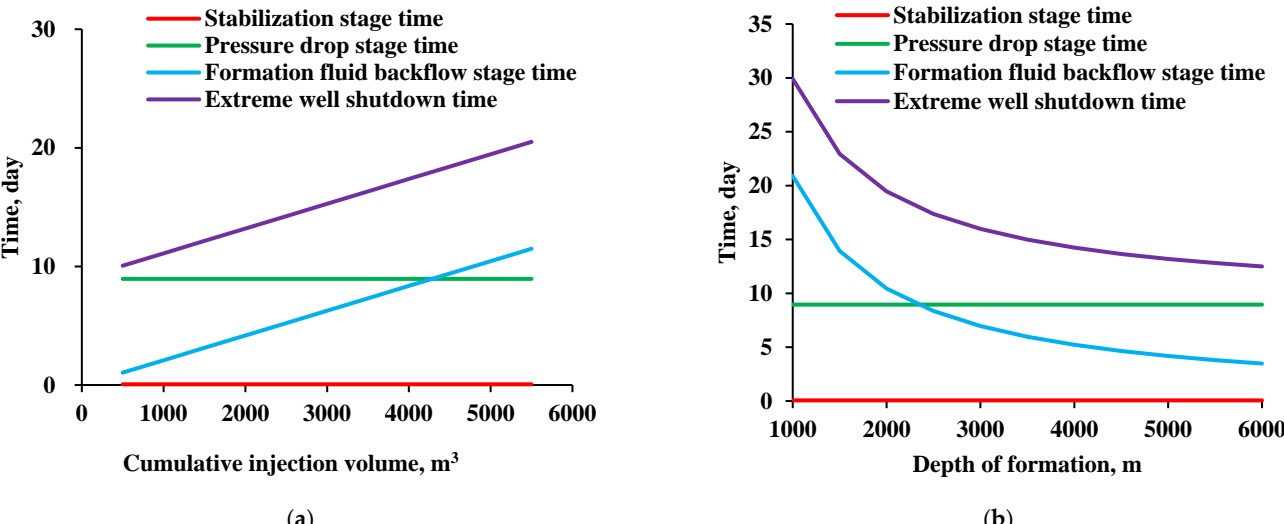

**Figure 7.** Influence of cumulative gas injection rate on the limit shut-in time of a waterproof compound. (**a**) Influence of accumulated gas volume on the limit shut-in time of a waterproof compound; (**b**) Influence of formation depth on the limit shut-in time of a waterproof compound.

It can be seen that the cumulative gas injection rate and formation depth have a greater impact on the ultimate shut-in time of the waterproof compound; the influence of the cumulative gas injection rate on the limit shut-in time of the waterproof compound is mainly reflected in the formation fluid backflow stage. With the increase in cumulative gas injection rate, the displacement distance increases, and the formation fluid backflow time becomes longer. The influence of formation depth on the limit shut-in time of the waterproof compound is mainly reflected in the formation fluid reflux stage. When the local formation depth is deep, the static pressure of the formation after shut-in is high, and the formation water reflux speed is slow.

## 4. Field Application

By the end of 2021, the alternative water and gas injection process facilities in S101 and the S16 well blocks had been completed, including one gas injection station and two injection distribution rooms. The S16 well block governs 32 gas injection wells, which can realize the alternate water and gas injection. Three injection allocation rooms are built in this well block. At present, there are 24 wells in the S101 and S16 pilot areas that have implemented water/gas alternate injection. Since 2015, 21 wells have been frozen and blocked. Most of the wells have been treated by injecting methanol, injection-plugging agents and other methods, and some of the wells have not been opened. At the wellhead, the injection pressure is 20~24 MPa and the injection temperature is about $-10$ °C. Due to the low formation permeability of the block, the design injection volume is 5~28 m$^3$/d. The size of the injection tubing is 48 mm and the well depth is about 2000 m. According to the field construction data, the statistics of freezing and plugging of $CO_2$ injection wells are shown in Table 3.

It can be seen from Table 3 that as of May 2021, 21 wells have been blocked by freezing, of which five have been blocked by hydrate during water injection; there is no hydrate blockage during gas injection, which is consistent with our research results. In addition, we compared the hydrate blockage after shut-in of actual gas injection wells with the model prediction results and found that the hydrate blockage time after shut-in of injection wells following water injection is relatively short. It is generally maintained between 54 and 86 h, and the relative error of simulation is 5.81–22.95%. Following the gas injection, the

plugging time of hydrate is relatively long after the shut-in of the injection well, which is basically maintained at about 1000 h. The minimum relative error of prediction is only 0.53%, which has a strong field application value.

**Table 3.** Hydrate blockage in the wellbore of carbon dioxide injection wells.

| Well No. | Construction Method in Case of Wellbore Hydrate Blockage | Wellhead Pressure (MPa) | Actual Wellbore Hydrate Plugging Time (h) | Simulation Prediction of Wellbore Hydrate Plugging Time (h) | Remarks |
|---|---|---|---|---|---|
| S96-T15 | Water flooding | 19 | 351 | - | The daily injection volume is 8.47 $m^3$/d |
| S64-53 | Water flooding | 23 | 286 | - | The daily injection volume is 13.2 v$m^3$/d |
| S64-56 | Water flooding | 23.2 | 458 | - | The daily injection volume is 7.12 $m^3$/d |
| S94-T16 | Water flooding | 18 | 437 | - | The daily injection volume is 9.24 $m^3$/d |
| S96-T13 | Water flooding | 19 | 514 | - | The daily injection volume is 12.17 $m^3$/d |
| S72-55 | Gas injection after well shut-in | 23.5 | 69 | 78 | The relative error of prediction is 13.04% |
| S72-58 | Gas injection after well shut-in | 23.5 | 68 | 76 | The relative error of prediction is 11.76% |
| S70-X58 | Gas injection after well shut-in | 18 | 74 | 85 | The relative error of prediction is 14.86% |
| S96-T12 | Gas injection after well shut-in | 23 | 57 | 71 | The relative error of prediction is 14.52% |
| S66-51 | Gas injection after well shut-in | 24 | 54 | 68 | The relative error of prediction is 15.25% |
| S66-57 | Gas injection after well shut-in | 22.2 | 86 | 91 | The relative error of prediction is 5.81% |
| S64-57 | Gas injection after well shut-in | 21.7 | 61 | 75 | The relative error of prediction is 22.95% |
| S66-51 | Gas injection after well shut-in | 24 | 84 | 93 | The relative error of prediction is 10.71% |
| S72-53 | Gas injection after well shut-in | 21.5 | 1029 | 968 | The relative error of prediction is 5.93% |
| S72-53 | Gas injection after well shut-in | 21.5 | 1147 | 1068 | The relative error of prediction is 6.89% |
| S68-51 | Gas injection after well shut-in | 23.5 | 964 | 752 | The relative error of prediction is 21.99% |
| S70-53 | Gas injection after well shut-in | 22.7 | 897 | 714 | The relative error of prediction is 20.40% |
| S68-52 | Gas injection after well shut-in | 21 | 1054 | 958 | The relative error of prediction is 9.11% |
| S72-54 | Gas injection after well shut-in | 21.5 | 1136 | 1130 | The relative error of prediction is 0.53% |
| S98-TX13 | Gas injection after well shut-in | 20 | 1367 | 1251 | The relative error of prediction is 8.49% |
| S96-T16 | Gas injection after well shut-in | 23 | 1259 | 1064 | The relative error of prediction is 15.49% |

## 5. Conclusions

Generally speaking, the freezing and plugging wells are mainly double-pipe injection wells and concentric pipe injection wells and are mainly affected by shut-in time or water

injection rate. The freezing blockage of the trunk line is accompanied by the freezing blockage of the shaft, which indicates that the freezing blockage of the shaft is the main reason for further freezing blockage of the trunk line. The main reasons for freezing and plugging of supercritical $CO_2$ water alternative injection wells are analyzed preliminarily and include long-term shutdown after alternative injection; improper operation when stopping injection; and starting and stopping of pumps. In the process of alternate injection, it is caused by three factors, one of which is slow injection speed. In the process of supercritical $CO_2$ water alternative injection, following injection the $CO_2$ in the formation will reverse-diffuse to the injection well end. With the continuous increase of daily water injection, the initial diffusion position and the time of $CO_2$ diffusion to the perforated hole after well shut-in gradually increase. The time of $CO_2$ reverse diffusion to the bottom of the well is 1.6–32.3 d, and the diffusion time in the perforated hole is 1.0–4.5 d. Therefore, the limit shut-in time for post-injection is 2.6–36.8 d. Following gas injection, the limit shut-in time of a waterproof compound can be divided into three stages according to the change of wellbore pressure: the pressure stabilization stage, pressure-drop stage and formation fluid return stage. The limit shut-in time of a waterproof compound is mainly affected by permeability, cumulative gas injection rate and formation depth following gas injection, and ranges from 20.0~30.0 d.

**Author Contributions:** Conceptualization, K.L.; methodology, G.C.; software, G.S.; validation, N.Z.; resources, X.L.; data curation, S.Z.; writing—original draft preparation, Y.B. All authors have read and agreed to the published version of the manuscript.

**Funding:** This research received no external funding.

**Conflicts of Interest:** The authors declare no conflict of interest.

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
