# Peer review of "Optimization of Anti-Plugging Working Parameters for Alternating Injection Wells of Carbon Dioxide and Water"

_processes, doi:10.3390/pr10112447_

Round 1
Reviewer 1 Report
The authors describe a scenario of CO2-water alternating flooding in which formation of gas hydrates is stated as a possible risk. If the reviewer understands right, as long as flooding fluid is injected there is no problem expected. As soon as the well is shut in, CO2 starts to diffuse towards the well and subsequently flows upwards towards the well head passing by a so-called freeze plugging section. If this is right, the authors shall find a more straight-forward explanation of the scenario. The explanations in the manuscript are complicated and repetitive.
The prevailing mass transport mechanisms are addressed properly but the authors lack understanding of the phase behavior of CO2 and even more of gas hydrates. It is recommended to have a look into the standard literature, such as “Sloan: Clathrate Hydrates of Natural Gases”.
A phase diagram of CO2 and of CO2-water is missing in which the described scenario shall be illustrated. The authors do not have a clear view on phase transitions and of the meaning of the supercritical state. This can be clarified with the respective phase envelope. The authors shall mention the conditions in terms of P and T in the scenario under consideration. CO2-gas hydrates only form below +10° C and at pressures in the gaseous or liquid state of CO2, not in the supercritical state. Gas hydrates normally occur in pipelines or wells close to the surface in cold regions or under subsea/deepsea conditions. The authors mention a temperature of 90° C in line 351, which is far beyond conditions of hydrate formation.
Author Response
Point 1: The authors describe a scenario of CO2-water alternating flooding in which formation of gas hydrates is stated as a possible risk. If the reviewer understands right, as long as flooding fluid is injected there is no problem expected. As soon as the well is shut in, CO2 starts to diffuse towards the well and subsequently flows upwards towards the well head passing by a so-called freeze plugging section. If this is right, the authors shall find a more straight-forward explanation of the scenario. The explanations in the manuscript are complicated and repetitive.
Response 1: For this problem, we have supplemented detailed explanations in the revised version of the paper. In the actual gas injection production process of the oilfield, the alternating injection wells of carbon dioxide and water will be frozen and blocked during water injection and shut in. Therefore, we have studied the normal shut in and water injection conditions respectively: for the frozen and blocked problem in the injection process, we have calculated the limit flow rate, Aimed at different shut in conditions (post gas injection conditions and post water injection conditions), targeted calculations were carried out.
Point 2: The prevailing mass transport mechanisms are addressed properly but the authors lack understanding of the phase behavior of CO2 and even more of gas hydrates. It is recommended to have a look into the standard literature, such as “Sloan: Clathrate Hydrates of Natural Gases”.
Response 2: Thank you very much for the suggestions of the reviewers. We have further improved the introduction of the paper according to the suggestions of the reviewers, added important references in recent years, and made targeted analysis.
Point 3: A phase diagram of CO2 and of CO2-water is missing in which the described scenario shall be illustrated. The authors do not have a clear view on phase transitions and of the meaning of the supercritical state. This can be clarified with the respective phase envelope. The authors shall mention the conditions in terms of P and T in the scenario under consideration. CO2-gas hydrates only form below +10℃ and at pressures in the gaseous or liquid state of CO2, not in the supercritical state. Gas hydrates normally occur in pipelines or wells close to the surface in cold regions or under subsea/deepsea conditions. The authors mention a temperature of 90℃ in line 351, which is far beyond conditions of hydrate formation.
Response 2: Thanks very much for the reminder of the reviewers. We have rectified the relevant contents of the paper according to the suggestions of the reviewers. The supercritical carbon dioxide mentioned in the paper refers to the state in the formation (temperature: 90℃, pressure: 30Mpa), and the freezing in the wellbore is mainly concentrated in a certain position of the injection well wellbore, which we have studied in detail before. (Reference:) In addition, we have supplemented and analyzed the phase diagram of carbon dioxide hydrate according to the suggestions of reviewers.

Reviewer 2 Report
In this manuscript, authors discussed the reasons for hydrate freezing and plugging in CO2 and water alternate injection wells. They clarified the distribution characteristics and sources of hydrate near the well, and established a coupling model to calculate the limit injection velocity and limit shut in time. The paper is within the scope of Processes, I recommend it to be published after revision. Below some comments to be addressed in a revision of the paper.
(1) The title is too long, I think it should be revised to be more refined. The title says “Cause Analysis of Freezing and Blocking”, but the there is no any analysis about blocking in the results part.
(2) Hydrate formation rate is crucial for the formation of freezing and blocking in supercritical CO2 and water alternate injection wells. It is necessary to discuss this content in the introduction part. The second paragraph should be divided to two paragraphs. There are too few references in this part, which should be enriched. And some recent investigation would be helpful (https://doi.org/10.1016/j.ijheatmasstransfer.2019.05.039).
(3) Figure 2 shows little useful information, a calculated result of bubble migration and distribution may be better.
(4) What method was used in the simulation? Commercial software or self programming? This should be expatiated clearly in the manuscript.
(5) The format of all Figures should be improved, the current version is rough.
(6) How to assure your calculation results are accurate? A comparison between the calculation results and experimental data or field data may be needed.
(7) A thorough check on the language use is needed to make the English writing more native.
Author Response
Point 1: The title is too long, I think it should be revised to be more refined. The title says “Cause Analysis of Freezing and Blocking”, but the there is no any analysis about blocking in the results part.
Response 1: According to the suggestions of reviewers, we have revised the title of the paper as follows: Optimization of Anti plugging Working Parameters for Alternating Injection Wells of Carbon Dioxide and Water
Point 2: Hydrate formation rate is crucial for the formation of freezing and blocking in supercritical CO2 and water alternate injection wells. It is necessary to discuss this content in the introduction part. The second paragraph should be divided to two paragraphs. There are too few references in this part, which should be enriched. And some recent investigation would be helpful (https://doi.org/10.1016/j.ijheatmasstransfer.2019.05.039).
Response 2: Thank you very much for the suggestions of the reviewers. We have further improved the introduction of the paper according to the suggestions of the reviewers, added important references in recent years, and made targeted analysis.
Point 3: Figure 2 shows little useful information, a calculated result of bubble migration and distribution may be better.
Response 3: Figure 3 is a further supplement to the rule of bubble migration. We have integrated Figure 2 to Figure 3 in the paper according to the suggestions of reviewers.
Point 4: What method was used in the simulation? Commercial software or self programming? This should be expatiated clearly in the manuscript.
Response 4: In this paper, different calculation methods are used for different working conditions, including finite element simulation method and theoretical derivation, which have been described in detail in this paper.
Point 5: The format of all Figures should be improved, the current version is rough.
Response 5: According to the suggestions of reviewers, we have modified all the pictures in the paper and adopted a tiff format greater than 300 dpi.
Point 6: How to assure your calculation results are accurate? A comparison between the calculation results and experimental data or field data may be needed.
Response 6: We have supplemented the field application at the end of the paper according to the suggestions of the reviewers.
Point 7: A thorough check on the language use is needed to make the English writing more native.
Response 7: We have polished the paper comprehensively to avoid many grammatical errors in the paper, so that readers can better understand our research.

Round 2
Reviewer 1 Report
The authors addressed the reviewers comments by showing the context of the work and mentioning operating conditions that are indeed within the hydrate formation region supporting the relevance of the work. The scientific quality of the work is average but it should be ok for publishing in this journal.
Author Response
We have refined the language in the paper and revised the references according to the suggestions of the reviewers.

Reviewer 2 Report
(1) The format of the fourth figure in Fig.1 should be improved.
(2) The references and citations in Introduction part are inconsistent in the current version.
Author Response
1.The fourth figure in Figure 1 has been modified according to the requirements of the reviewer.
2.According to the requirements of the reviewer, the reference position in the text has been revised.